# Repeated Intravenous Administration of Human Neural Stem Cells Producing Choline Acetyltransferase Exerts Anti-Aging Effects in Male F344 Rats

**DOI:** 10.3390/cells12232711

**Published:** 2023-11-26

**Authors:** Jangbeen Kyung, Dajeong Kim, Kyungha Shin, Dongsun Park, Soon-Cheol Hong, Tae Myoung Kim, Ehn-Kyoung Choi, Yun-Bae Kim

**Affiliations:** 1College of Veterinary Medicine, Chungbuk National University, Cheongju 28644, Republic of Korea; 2Department of Biology Education, Korea National University of Education, Cheongju 28173, Republic of Korea; 3Department of Obstetrics and Gynecology, Korea University College of Medicine, Seoul 02841, Republic of Korea; 4Central Research Institute, Designed Cells Co., Ltd., Cheongju 28576, Republic of Korea

**Keywords:** aging, cognitive function, physical activity, human neural stem cell, choline acetyltransferase, acetylcholine, growth factor, neurotrophic factor

## Abstract

Major features of aging might be progressive decreases in cognitive function and physical activity, in addition to withered appearance. Previously, we reported that the intracerebroventricular injection of human neural stem cells (NSCs named F3) encoded the choline acetyltransferase gene (F3.ChAT). The cells secreted acetylcholine and growth factors (GFs) and neurotrophic factors (NFs), thereby improving learning and memory function as well as the physical activity of aged animals. In this study, F344 rats (10 months old) were intravenously transplanted with F3 or F3.ChAT NSCs (1 × 10^6^ cells) once a month to the 21st month of age. Their physical activity and cognitive function were investigated, and brain acetylcholine (ACh) and cholinergic and dopaminergic system markers were analyzed. Neuroprotective and neuroregenerative activities of stem cells were also confirmed by analyzing oxidative damages, neuronal skeletal protein, angiogenesis, brain and muscle weights, and proliferating host stem cells. Stem cells markedly improved both cognitive and physical functions, in parallel with the elevation in ACh levels in cerebrospinal fluid and muscles, in which F3.ChAT cells were more effective than F3 parental cells. Stem cell transplantation downregulated CCL11 and recovered GFs and NFs in the brain, leading to restoration of microtubule-associated protein 2 as well as functional markers of cholinergic and dopaminergic systems, along with neovascularization. Stem cells also restored muscular GFs and NFs, resulting in increased angiogenesis and muscle mass. In addition, stem cells enhanced antioxidative capacity, attenuating oxidative damage to the brain and muscles. The results indicate that NSCs encoding ChAT improve cognitive function and physical activity of aging animals by protecting and recovering functions of multiple organs, including cholinergic and dopaminergic systems, as well as muscles from oxidative injuries through secretion of ACh and GFs/NFs, increased antioxidant elements, and enhanced blood flow.

## 1. Introduction

Throughout history, the elongation of lifespan has been one of the hot issues in all animals on earth, not limited to humans. It is well documented that the SIRT1 gene plays a central role in the regulation of many genes, including NF-κB mediating inflammation, as well as mTOR protecting cells and organs against damage factors [1]. For example, resveratrol was found to exert health-enhancing and lifespan-extending properties both in vitro and in vivo in a number of model organisms via the SIRT1 gene [1,2]. The anti-aging effects of calorie restriction were also mediated by mTOR-related metabolic and immune regulations [3,4]. In our super-aging society, however, health span rather than lifespan might be more important since incurable degenerative diseases threaten our healthy lives. During the process of aging, various visual and functional changes in the body appear. The main functional changes in aging might be impairments of cognitive function, including learning and memory performances, as well as physical function, such as movement capacity.

It was demonstrated that brain aging in animals resulted in precipitous decreases in NSCs and neurogenesis [5,6]. In such a microenvironment, neurogenesis may not be enough for the replacement of dead cells, and the neural network is changed insufficiently for proper signal transduction. These changes lead to cognitive impairment focusing on learning and memory functions [6,7]. It is known that degeneration of the cholinergic system governing memory processing leads to cognitive dysfunction. The concentration of ACh, a major neurotransmitter responsible for memory acquisition, was diminished in aged animals as a result of cholinergic nerve degeneration [8,9,10]. ACh is synthesized by ChAT and released to synapses via vesicular acetylcholine transporter (VAChT). Therefore, degenerative changes in cholinergic nerves in aged persons and Alzheimer’s disease (AD) patients result in learning and memory loss following the disintegration of the cholinergic factors, including ChAT [9,10,11,12,13].

Another prominent phenomenon of aging may be decreased physical activity. Such aggravations in physical functions are, in part, mediated by reduced dopaminergic regulation [14,15]. Physical activity is significantly reduced in Parkinson’s disease (PD) patients and PD model animals with dopaminergic system impairment. The dopaminergic nervous system functions mainly via D2 receptors (D2R) distributed in the cortical region, and the cortex is also a major site for motor neuron networking [16]. Moreover, cholinergic system degeneration during aging lowers dopaminergic activity via its regulation procedures. It was confirmed that decreased ACh synthesis negatively affected the dopaminergic system, diminishing physical activity and movement [17]. Among neurotrophins, glial cell-derived neurotrophic factor (GDNF) regulates dopaminergic nerves by raising monoamine production. Collectively, physical activity is mainly controlled by the regulation of dopaminergic and cholinergic nervous systems [18]. In addition to the control by the central nervous system, physical stamina depends on muscular strength and endurance to fatigue. In aged rats, muscle mass and strength declined along with a decreased distribution of blood vessels in the muscles [19].

Despite many research reports on the palliative improvement in cognitive and physical functions using small molecule drugs and natural products, potential candidates showing long-term salient effects are not available. Although fasting induced the rejuvenation of stem cells in invertebrate *Schmidtea mediterranea* [20], it is too hard to reproduce such an effect in mammals. Instead, there are some reports on the anti-aging activities of stem cells. NSC transplantation enhanced learning and memory function in aged animals [21]. Adipose-derived stem cells (ADSCs) enhanced stamina not only by attenuating tissue injury but also by strengthening the muscles via the production of brain-derived neurotrophic factor (BDNF) [22]. In addition, we established human NSCs encoding the ChAT gene (named F3.ChAT cells), an enzyme responsible for ACh synthesis, by inserting the ChAT gene into the F3 NSC line. Single intracerebroventricular (ICV) transplantation of F3.ChAT cells increased brain ACh levels and thereby fully restored the learning and memory functions of AD model animals [23,24]. Indeed, a single injection of the F3.ChAT cells also markedly recovered both the cognitive function and physical activity of aged mice by increasing the levels of ACh and various GFs/NFs [25].

Based on the previous results, we investigated the anti-aging effects of F3 and F3.ChAT NSCs focusing on cognitive dysfunction and physical inactivation as well as underlying action mechanisms in naturally aging rats following repeated life-long intravenous (IV) injections from 10 months of age.

## 2. Materials and Methods

### 2.1. Human NSC Line

Human NSCs (F3 and F3.ChAT) were provided by Prof. SU Kim at the University of British Columbia [23,24,26,27]. The cells were cultivated in Dulbecco’s Modified Eagle’s Medium (DMEM) supplemented with 10% fetal bovine serum and penicillin/streptomycin (Invitrogen, Carlsbad, CA, USA) in a 5% CO_2_ incubator.

### 2.2. Aging Animals and NSC Transplantation

Ten-month-old Fisher 344 rats were from DBL (Eumseong, Republic of Korea). The animals were housed in a room with constant temperature (23 ± 2 °C), relative humidity (55 ± 10%), and 12-h light cycle and fed on standard rodent chow and purified water. Animals were injected IV with NSCs (1 × 10^6^ cells/rat) in 100 μL saline. The rats (*n* = 20/group), excepting control (Young and Old) groups, received the cells once a month repeatedly for 11 months (from 11 to 21 months of age).

### 2.3. Measurement of Physical Activity

#### 2.3.1. Locomotor Activity

Global activities, that is, spontaneous activity and exploratory behaviors, were monitored via a video tracking system (Smart v2.5; Panlab Technology, Barcelona, Spain), connected to a CCTV monitor (Samsung, Changwon, Republic of Korea) 1 week after cell administration. After placing rats in a quiet chamber with dim light, the types of movement, i.e., resting (<200 cm/s), slow-moving (200–500 cm/s), and fast-moving (≥500 cm/s) times, were recorded for 5 min, and the ratio was analyzed.

#### 2.3.2. Rota-rod Performance

Motor balance and coordination were evaluated using a rota-rod test system (Panlab Technology) 1 week after stem cell injection. Rats were placed on a rotating (12 rpm) rod, and the fall-off time was recorded. The average latency was calculated from 3 replicates.

### 2.4. Measurement of Learning and Memory Functions

#### 2.4.1. Passive Avoidance Performance

In order to assess memory acquisition and retention, rats were subjected to passive avoidance trials in a Shuttle box (ENV-010MD; Med Associates Inc., St. Albans, VT, USA) with light and dark chambers when the survival rate of animals was 80% (21 months old). In the trials, electric shock (1 mA for 2 s) was delivered when animals entered the dark chamber from the light room. The latency time of staying in the light chamber was recorded in each 7-day trial, once a day. The end-point was set to 300 s, denoting full acquisition of memory.

#### 2.4.2. Water Maze Performance

The spatial memory of rats was evaluated through Morris water maze performance. A round water bath (180 cm in diameter) filled with water (22 ± 2 °C) was divided into four quadrants and equipped with a submerged platform (10 cm in diameter). The animals were subjected to 7-day trials to find the platform hidden by white styrofoam granules (5 cm in diameter) on the surface of the water. Escape latency time spent to escape onto the platform in each trial was recorded. The end-point was set to 300 s.

### 2.5. Analysis of ACh Concentration

The animals were sacrificed 24 h after the final learning/memory tests. Cerebrospinal fluid (CSF) was collected, and gastrocnemius muscles were obtained and homogenized to analyze ACh concentration. ACh concentrations in CSF and muscle homogenate were measured with an Amplex Red acetylcholine/acetylcholinesterase assay kit (Molecular Probes, Eugene, OR, USA) according to the manufacturer’s instructions. The produced resorufin fluorescence was recorded in a microplate reader using excitation in the range of 530–560 nm and emission at 590 nm.

### 2.6. Stem Cell Distribution

The brains were perfusion-fixed with a paraformaldehyde solution, followed by cryoprotection in 30% sucrose for 3 days to confirm the distribution of transplanted F3 and F3.ChAT cells. Coronal cryosections (30 μm in thickness) were prepared and immunostained with an antibody of human mitochondria (hMito, 1:200; mouse monoclonal, Chemicon, Temecula, CA, USA). In order to identify proliferating NSCs, brain sections were immunostained with antibodies of nestin (1:200; mouse polyclonal, Chemicon) and Ki-67 (1:200; rabbit polyclonal, Chemicon). Brain sections were incubated with primary antibodies overnight at 4 °C, followed by secondary antibodies conjugated with Alexa Fluor-488 or −594 (1:500; rabbit polyclonal, Molecular Probes) for 2 h at room temperature. The images were photographed with a laser-scanning confocal microscope (LSM710; Zeiss, Oberkochen, Germany) and merged.

### 2.7. RT-PCR Analysis

Total RNA was extracted from the brain homogenate using TRIzol (Invitrogen). Complimentary DNA templates were prepared from 1 μg of total RNA primed with oligodT primers using 40 U of Moloney Murine Leukemia Virus reverse transcriptase (Promega, Madison, WI, USA), followed by 40 PCR cycles. RT-PCR products were separated through 1.2% agarose gel electrophoresis. The primers used for RT-PCR (ChAT, VAChT, ChT1, m1-AChR, nAChR α5, nAChR 2β, AChE, TH, VMAT2, DAT, D1R, D2R, CCL11, and GAPDH) are listed in Appendix A (Bioneer, Daejeon, Republic of Korea).

### 2.8. Western Blot Analysis

Whole brains and gastrocnemius muscles of animals were homogenized in RIPA buffer (Sigma-Aldrich, St. Louis, MO, USA). Proteins were quantified with a BCA Protein Assay kit (Pierce, Rockford, IL, USA). After heat-denaturation, proteins were separated by electrophoresis on 7.5% SDS-polyacrylamide gels and transferred to a polyvinylidene difluoride membrane. The membrane was incubated with antibodies specific to microtubule-associated protein 2 (MAP2) (1:500; rabbit polyclonal, Santa Cruz Biotechnology, Santa Cruz, CA, USA) overnight at 4 °C, followed by a secondary goat anti-rabbit IgG conjugated with horseradish peroxidase (1:2000; Santa Cruz Biotechnology) for 2 h at room temperature. The membrane was then developed using an enhanced chemiluminescence solution (Pierce, Rockford, IL, USA).

### 2.9. Enzyme-Linked Immunosorbent Assay (ELISA)

The major GFs and NFs related to tissue protection, regeneration, and angiogenesis were determined via ELISA, according to the manufacturer’s instructions. Briefly, brain or gastrocnemius muscle homogenate was put into the ELISA wells with antibodies specific for BDNF (ab213899; Abcam), nerve growth factor (NGF, ab193736; Abcam), GDNF (ab213901; Abcam), vascular endothelial growth factor (VEGF, ab100786; Abcam), or insulin-like growth factor-1 (IGF-1, ab213902; Abcam), and incubated for 1 h at room temperature. After washing 3–4 times with wash buffer, the primary antibodies were added and reacted for 1 h at room temperature and then washed again. Secondary antibodies were treated and incubated for 30 min, and substrates were applied for 20 min to develop color. After stopping the color development with a stop solution, the absorbance was measured at 450 nm.

### 2.10. Identification of Capillary

Microvessel density was analyzed by immunostaining von Willebrand Factor (vWF) to identify endothelial cells. The brain and gastrocnemius muscles were fixed in a paraformaldehyde solution, and paraffin-embedded sections were prepared. After being pretreated with citrate buffer (pH 6.0), the sections were incubated with primary antibody of vWF (1:200; rabbit polyclonal, Chemicon) overnight at 4 °C, followed by biotinylated secondary antibody for 1 h at room temperature. After treatment with the avidin–biotin complex kit (Vector Laboratory, Burlingame, CA, USA), the tissues were developed with diaminobenzidine (Sigma-Aldrich). Sections were observed under the field of ×200 of a light microscope.

### 2.11. Antioxidative Activity

To assess oxidative tissue injury, brain and gastrocnemius muscles were homogenized in 9 volumes of 10 mM sodium phosphate-buffered saline (PBS, pH 7.4) at 4 °C. The homogenate was acidified to pH 3.5 by adding SDS (8.1% solution) and 20% acetic acid. After adding 2-thiobarbituric acid (TBA, 0.75% solution), the mixture was boiled in a glass tube capped for 30 min at 95 °C. Samples were cooled on ice, centrifuged at 13,000× *g* for 10 min, and absorbance of the supernatant was read at 532 nm for the quantification of thiobarbituric acid-reactive substances (TBARS).

### 2.12. Statistical Analysis

Data are presented as mean ± standard deviation. Statistical analysis was performed with SPSS version 26.0 program (SPSS Inc., Chicago, IL, USA). Differences among groups were analyzed with one-way ANOVA, followed by Tukey’s HSD at a level of *p* < 0.05.

## 3. Results

### 3.1. Improvement in Physical Activity

Old animals (11–20 months old) exhibited severely decreased movement as measured via locomotor activity (Figure 1A). The resting time of aged rats markedly increased to 65–80%, while young animals displayed 70–95% moving (slow- and fast-moving) activities. However, transplantation of F3 or F3.ChAT cells significantly improved the activity of old animals, showing increased slow-moving time up to 20 months of age when 20% of the rats died, although fast-moving time increased at 11 months following F3.ChAT treatment.

The latency time in the rota-rod performance of aged animals also markedly decreased to 10–40% of that of young animals, and especially, the impairment of performance further increased according to the progress of aging (Figure 1B). Notably, repeated transplantation of F3 or F3.ChAT cells greatly improved the motor coordination of old rats, in which F3.ChAT was more effective to some extent than F3 cells.

### 3.2. Recovery of Cognitive Function and ACh Concentration

In addition to the loss of physical activity, 21-month-old rats showed severe impairment of cognitive function (Figure 2A,B). The aged rats showed a delayed increase in the latency time in spite of repeated trials in the passive avoidance performance, by comparison with the full memory acquisition at the fifth trial in young animals (Figure 2A). In water maze performance (Figure 2B), the old animals did not show memory-acquiring capacity during seven repeated trials, in contrast to the rapid decrease in the escape latency in young animals. However, repeated transplantation of F3 cells significantly improved the cognitive function in both performances. Notably, higher cognition-enhancing effects were achieved with the transplantation of F3.ChAT cells, rather than F3 cells, in both early (3–4 days) passive avoidance and water maze performances.

The ACh concentration in CSF (4.3 μM) of aged (21-month-old) rats was much lower than that of young (7-week-old) animals (7.3 μM) (Figure 2C). Such decreased ACh level in CSF was significantly restored to 5.1 and 6.1 μM following transplantation of F3 and F3.ChAT cells, respectively. Muscular ACh concentration in old rats (1.6 μmole/g tissue) was significantly lower than that in young animals (2.4 μmole/g tissue) (Figure 2D). Muscular ACh level was also recovered to 2.0 and 2.5 μmole/g tissue by transplantation of F3 and F3.ChAT cells, respectively. Notably, F3.ChAT cells were more effective than F3 cells in the recovery of brain and muscular ACh levels.

### 3.3. Distribution of Transplanted Cells

hMito-immunoreactivity was detected in the brain after 12-times transplantation of F3 cells (1 × 10^6^ cells/rat) into 10-month-old animals (Figure 3A). The transplanted cells were distributed predominantly in the hippocampus (89.1 cells/mm^2^), although a part of the cells were found in the cortex (34.2 cells/mm^2^) (Figure 3B). Transplanted F3.ChAT cells were also detected in the hippocampus (103.7 cells/mm^2^) and cortex (48.0 cells/mm^2^) of rats (Figure 3A,C), indicative of higher penetration of F3.ChAT cells into the brain than F3 cells.

### 3.4. Cholinergic and Dopaminergic Activation in the Host Brain

Expressions of functional genes of the cholinergic system were markedly changed according to aging. That is, genes for synthesis, secretion, and reception of Ach, including ChAT, VAChT, ChT1, m1-AChR, nAChR α5, and nAChR β2 decreased, while the gene of AChE, an ACh-degrading enzyme, increased in aged animals (Figure 4A). Such alterations in the gene expression of cholinergic markers were remarkably restored after transplantation of F3 or F3.ChAT cells, in which F3. ChAT cells were superior to F3 cells.

Expressions of genes of the dopaminergic system, such as TH, VMAT2, DAT, D1R, and D2R, also decreased in aged animals. However, the decreased expression of genes associated with dopamine synthesis, transportation, and reception were markedly recovered following the transplantation of F3 or F3.ChAT cells, in which F3.ChAT cells were more effective than F3 cells in D1R and D2R restorations (Figure 4B).

### 3.5. Restoration of Brain and Muscular GFs and NFs

Concentrations of GFs and NFs, including BDNF, NGF, GDNF, and VEGF, related to neuroprotection and neuroregeneration in the brain significantly decreased in aged animals compared with the levels in young animals (Figure 5A). However, these GFs/NFs were upregulated by transplantation of F3 or F3.ChAT cells, in which F3.ChAT cells were superior to F3 parental cells. Moreover, F3.ChAT cells increased VEGF related to angiogenesis up to twice the level in young rats.

Muscular levels of BDNF involved in muscle growth and myofiber differentiation decreased in aged animals and were markedly restored by transplantation of F3 or F3.ChAT cells (Figure 5B). In addition, VEGF related to angiogenesis was considerably recovered following treatment with the cells. Notably, GDNF and IGF-1 related to motor neuron development and myogenesis, respectively, were greatly upregulated higher than the levels in young animals.

### 3.6. Angiogenic Effects

In old rats, the vWF-positive microvessel density in the brain was reduced to 27.8% of young animals (Figure 6A,B), and the relative brain weight decreased, meaning brain atrophy (Figure 6C). Interestingly, transplantation of F3 or F3.ChAT cells led to an increase in the number of blood vessels, indicative of preservation or neovascularization (Figure 6B), in which F3.ChAT was much more effective than F3 cells. Such effects may be related to the attenuated brain atrophy (Figure 6C).

In gastrocnemius muscles of aged animals, the vessel density also decreased to 66.7% of young animals (Figure 6D,E), resulting in a significant decrease in muscle weight (Figure 6F). Treatment with F3 or F3.ChAT cells increased the number of capillaries and muscle mass, although the effects were more prominent in F3.ChAT cells than F3 cells (Figure 6E,F).

### 3.7. Antioxidative Effects

The concentrations of TBARS, by-products of lipid peroxidation, significantly increased in the brain and muscles of old rats compared with those of young animals (Figure 7). Such increased oxidative damage was nearly fully attenuated by transplantation of F3 or F3.ChAT cells.

### 3.8. Neuroprotection and Neuroregeneration

The number of host neural (nestin-positive) stem cells in old rats markedly reduced to 26.9% of young animals (Figure 8A,B). Most of the nestin-positive cells also exhibited immunoreactivity to Ki-67, a proliferating cell marker, indicating that the host neural stem cells are actively proliferating. Interestingly, transplantation of F3 or F3.ChAT cells significantly increased the number of host proliferating (Ki-67-positive) cells, in which F3.ChAT cells were more effective than F3 cells (Figure 8B).

On the other hand, the expression of CCL11, an inhibitor of neurogenesis, increased during aging, in parallel with the marked decrease in the content of MAP2, a neuronal skeletal protein (Figure 8C). The changes in CCL11 expression and MAP2 production were fully reversed following transplantation of F3 or F3.ChAT cells.

## 4. Discussion

In this research, F3 and F3.ChAT NSCs exhibited anti-aging effects against spontaneous systemic aging. As a result of the anti-aging effect of transplanted cells, cognitive function was improved, in which both passive avoidance and water maze performances were markedly enhanced by F3 NSCs. Interestingly, the cognition-enhancing effect of F3.ChAT cells were much higher than that of F3 parental cells, exerting full memory acquisition within the fifth–seventh trials comparable to young animals.

It is well known that the acquisition of memory is mainly regulated by ACh of the cholinergic nervous system. According to the degenerative change in the cholinergic system during aging, learning and memory functions are gradually inactivated [8,28]. In PCR analysis, the expression of genes for ACh synthesis and release decreased in aged animals, whereas AChE, an ACh-degrading enzyme, increased. F3 and F3.ChAT NSCs reversed the changes in the gene expression related to both ACh synthesis and degradation, normalizing cholinergic function, as confirmed by recovery of ACh concentration. The gene expression of muscarinic and nicotinic receptors of ACh, responsible for memory acquisition, was recovered similarly. Notably, such effects on the reactivation of the cholinergic system, i.e., gene expression of cholinergic system markers as well as ACh concentration, were outstanding in the rats transplanted with F3.ChAT cells. In previous reports, F3.ChAT cells were also superior to their parental F3 cells in the recovery of cognitive function of AD model rats and aged mice, which is indicative of the major role of ChAT expression [23,24,25]. Interestingly, the cholinergic features were specifically affected by mitochondrial tRNA fragments in female human AD patients but not in males [29]. So, it remained to clarify comparative changes in the tRNA fragments between female and male rats as an important factor during aging and to assess the gender-different effects of stem cell therapy, even though there may be additional differences between humans and rodents.

From the immunostaining on hMito, it was confirmed that many NSCs entered the aging brain after repeated IV injections. Transplanted cells were observed predominantly in the hippocampus and cortex. The permeability of the blood–brain barrier (BBB) increased during aging, which might be mediated by oxidative damage [30]. Thus, it is believed that IV-treated cells migrated passing BBB into the aged brain and contributed to the increased brain ACh level, as was also observed previously in AD model and aging animals [23,24,25]. Muscarinic and nicotinic receptors are widely distributed in the hippocampus and cortex and play central roles in spatial and working memories [31,32]. ChAT is also one of the important components of motor neurons [33]. Repeated IV injection of NSCs markedly recovered the decreased MAP2 and cholinergic and dopaminergic markers in aged rats, indicative of a structural restoration of brain integrity. The improvements in cognitive and motor functions following F3.ChAT cell transplantation might be due to the increased ACh levels in the brain and muscles originating from the transplanted stem cells and restored host cholinergic neurons. In addition, the restored dopaminergic nervous system is believed to additionally improve physical activity.

Furthermore, neurogenesis plays a key role in the retention of learning and memory. Neurogenesis was promoted by neural progenitor cells’ maturation, and cognitive function was improved by an increase in neural cell population [28,34,35]. Cognitive function that declined with aging was recovered with neurogenesis promotion near the hippocampus, which is the main region associated with cognitive function. Notably, NSCs enhanced learning and memory function in aged animals [21]. In the present study, F3 and F3.ChAT NSCs also promoted neurogenesis and inhibited neuronal degeneration, i.e., stem cell transplantation induced proliferation of host nestin-positive cells mainly in the hippocampus, as confirmed by immunoreaction to Ki-67, wherein F3.ChAT cells were more effective than F3 parental cells. Such a neurogenic effect of stem cells might be mediated by GFs/NFs, such as BDNF and NGF [36,37]. BDNF and NGF promoted neurogenesis, proliferation, and differentiation into neurons in the hippocampus [38,39]. Indeed, BDNF activated the cholinergic nervous system by upregulating ChAT expression [38,39,40]. Similarly, NGF enhanced the survival of hippocampal neurons and regulated innervation and nerve terminal signaling [41,42,43].

In ELISA, these factors were remarkably recovered F3- or F3.ChAT-treated animals, although the effect of F3 cells was lower than F3.ChAT cells. In our previous reports, F3.ChAT cells expressed and produced diverse neurotrophins [44], and transplantation of the cells restored GFs and NFs levels in the brain tissue [25,37,44,45]. Therefore, it is suggested that the GFs/NFs might have contributed to the neurogenesis of the aging animals.

Brain angiogenesis is an important factor in neurogenesis. Reduced angiogenic activity in the aged brain was mostly related to the reduction in VEGF expression in response to degenerative change [46]. As a consequence, impaired VEGF production hinders neurogenesis, resulting in a decrease in neuronal population [46,47]. In stroke or ischemic injury, angiogenesis was promoted, which might have been mediated by GFs, including VEGF [48,49]. Interestingly, the microvessel density decreased in aged rat brains and was remarkably restored by F3.ChAT cell treatment. In parallel with the angiogenic activity, F3. ChAT cells markedly increased the VEGF concentration in the brain, while F3 cells did not increase both the VEFG production and vessel density. F3.ChAT cells also highly expressed VEGF and increased the capillary density in middle cerebral artery occlusion stroke model animals [44]. Accordingly, the attenuation of brain atrophy and restoration of muscle mass may be mediated in part by microvessel preservation and possible neovascularization. Notably, the expression of CCL11, a chemokine inhibiting neuroregeneration, significantly increased in aged animals [50]. However, the CCL11 expression was fully suppressed by F3 or F3.ChAT cells. As a consequence of such neurogenic effects of F3 and F3.ChAT cells, it is believed that host neural (nestin-positive) stem cells underwent proliferation (Ki-67-positive), and thereby neuronal skeletal protein MAP2 was recovered following NSCs transplantation [44,45,51].

In addition to the enhancement of ACh synthesis by cholinergic system activation, the dopaminergic system was also activated by F3 and F3.ChAT NSCs. That is, F3 or F3.ChAT cell transplantation recovered the expression of several genes related to dopamine synthesis and release, including TH, VMAT2, and DAT. Ideally, expression of dopaminergic receptors such as dopamine receptors 1 and 2 were recovered. Such effects were higher in F3.ChAT-treated rats’ brains than in F3-treated animals. As supported, dopaminergic activation was regulated by the cholinergic nervous system with the secretion of NFs such as GDNF [44,52], and dopaminergic activation enhanced physical activity and extended moving distance in F344 rats [14,53]. As a result of dopaminergic activation, physical activities (moving times in locomotor activity and rota-rod performance) increased following F3.ChAT cell transplantation.

Related to muscular differentiation and innervation, physical activities were regulated by GFs/NFs. In particular, BDNF, IGF, and GDNF facilitated muscle growth, myofiber differentiation, axonal sprouting, muscle innervation, synaptic plasticity, and ACh receptor expression in the neuromuscular junction [19,54,55,56,57]. Therefore, such GFs and NFs increased by stem cell administration might have led to the muscular innervation and activation of cholinergic and dopaminergic nerves, as observed in our study demonstrating a major role of BDNF in the stamina-enhancing effect of ADSCs [22]. In addition, VEGF enhanced muscular angiogenesis for sufficient glucose and oxygen supplies [58,59]. Notably, F3.ChAT cells recovered the microvessel density in the muscles of aged rats, leading to an increase in muscle mass as well as attenuation of brain atrophy.

Although cognitive function and physical activity are regulated by the nervous system and muscular integrity, such regulatory systems are impaired following tissue injuries from oxidative stress, a typical consequence of the aging process. Oxidative damage of mitochondrial DNA induces neuronal degeneration, followed by glial cell response [60]. Interestingly, GFs/NFs, including BDNF, NGF, VEGF, and IGF, possessing antioxidative activities, decreased in the aged brain and muscles [61,62], indicative of the multiple roles of GFs/NFs in protecting tissues and organs related to the antioxidative defense system. F3 and F3.ChAT cells protected against oxidative injury of muscles and brain tissues. Therefore, it is explained that the increased physical activity of aged rats treated with stem cells resulted not only from enhanced cholinergic and dopaminergic control but also from improved muscular integrity.

Actually, we developed F3.ChAT human NSCs for the therapy of human Alzheimer’s disease (AD). The cells can be used for human patients after approval of health authorities, including the Korean Food and Drug Administration. In fact, repeated injection of human cells may induce immune responses in animals and thereby can cause shock or be eliminated by antibodies. In our previous studies, however, the F3.ChAT cells induced negligible immune responses and, especially, did not show remarkable adverse effects, including antigenicity (immunotoxicity) in non-clinical safety tests (data not shown). Therefore, it is expected that the efficacies of human NSCs in human patients may be at least higher than in rodents.

Taken together, it is believed that the anti-aging effects of F3 and F3.ChAT cells were mediated by combinational activities on ACh production, neuroprotection and neuroregeneration, cholinergic and dopaminergic system activation, antioxidation and angiogenesis, and muscular regeneration accompanying GFs/NFs secretion. The anti-aging effects obtained with F3.ChAT cells were superior to those of F3 cells, especially as observed in the improvement in learning/memory performance and moving behavior. Therefore, it is suggested that human NSCs over-expressing the ChAT gene could be a candidate to delay the aging process characterized by cognitive dysfunction and physical activity impairments.

## Figures and Tables

**Figure 1 cells-12-02711-f001:**
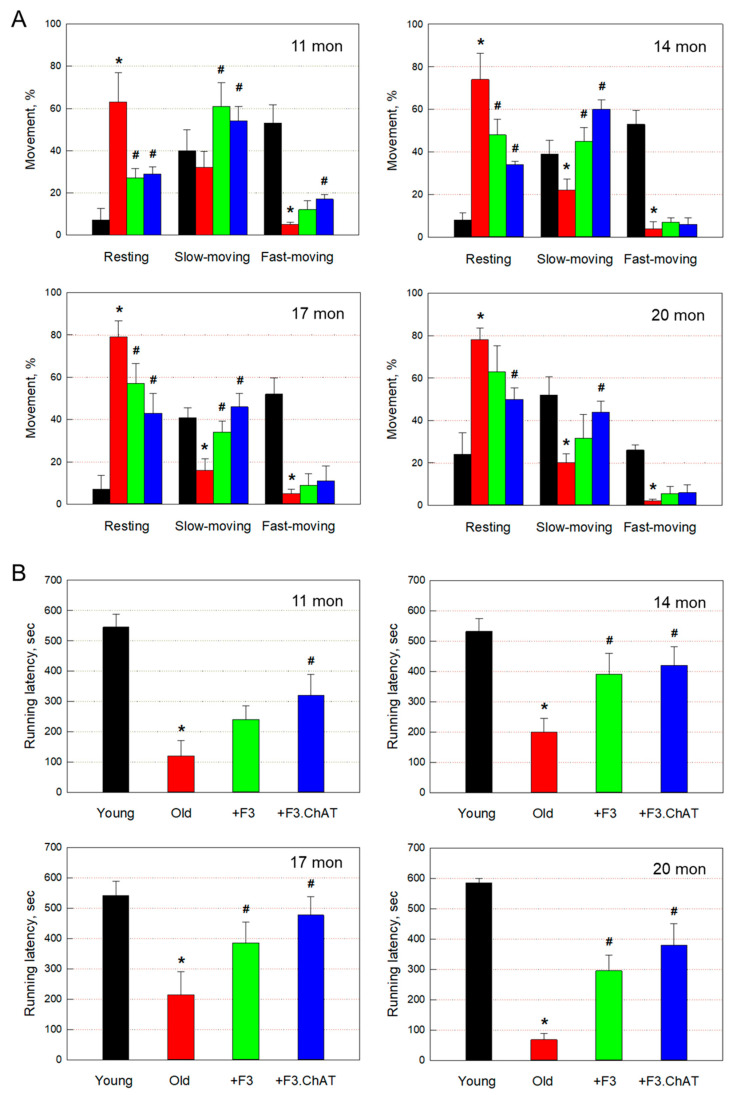
Locomotor activity (resting, slow-moving, and fast-moving times) ((**A**), *n* = 8–10/group) and rota-rod performance ((**B**), *n* = 8–10/group) of rats at 11–20 months of age. Black: young animals (7 weeks old); red: aged animals; green: aged animals transplanted with F3 cells (1 × 10^6^ cells every month); blue: aged animals transplanted with F3.ChAT cells (1 × 10^6^ cells every month). * Significantly different from young animals (*p* < 0.05). # Significantly different from aged animals (*p* < 0.05).

**Figure 2 cells-12-02711-f002:**
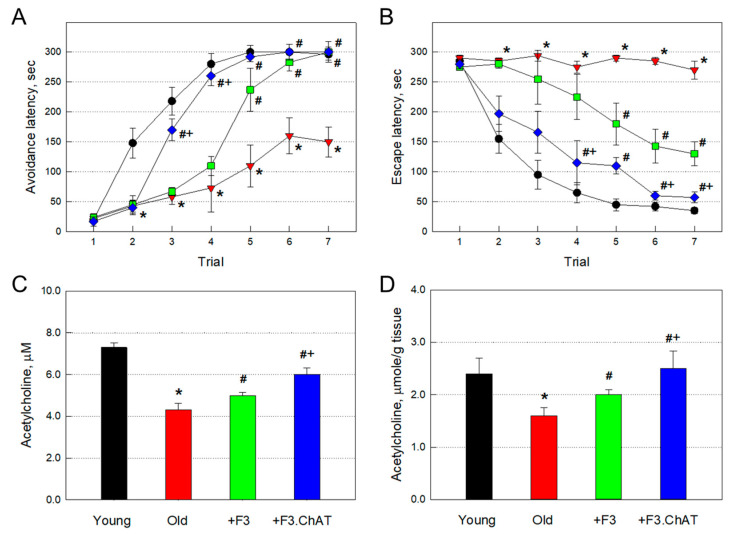
Passive avoidance (**A**) and Morris water maze (**B**) performances (*n* = 8–10/group) and acetylcholine levels (*n* = 10/group) in the brain (**C**) and muscles (**D**) at 21 months of age. Black: young animals (7 weeks old); red: aged animals; green: aged animals transplanted with F3 cells (1 × 10^6^ cells every month); blue: aged animals transplanted with F3.ChAT cells (1 × 10^6^ cells every month). * Significantly different from young animals (*p* < 0.05). # Significantly different from aged animals (*p* < 0.05). + Significantly different from F3 treatment (*p* < 0.05).

**Figure 3 cells-12-02711-f003:**
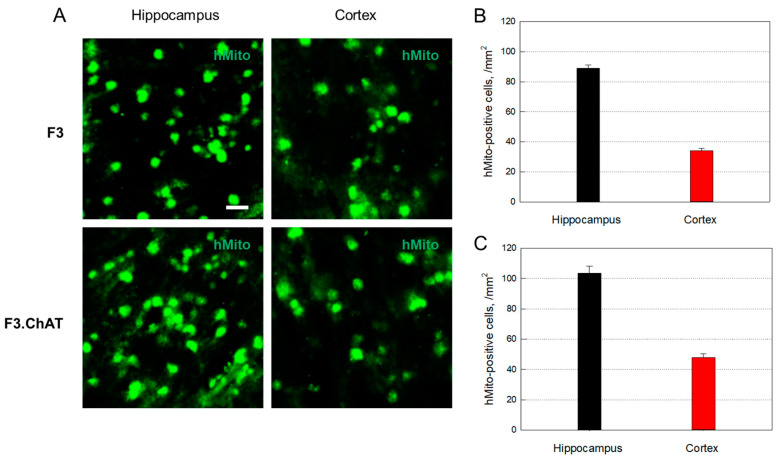
Distribution of the transplanted human (hMito-positive) F3 ((**A**,**B**), *n* = 8–10/group) and F3.ChAT ((**A**,**C**), *n* = 8–10/group) cells in the brain tissues. Sale bar = 20 μm.

**Figure 4 cells-12-02711-f004:**
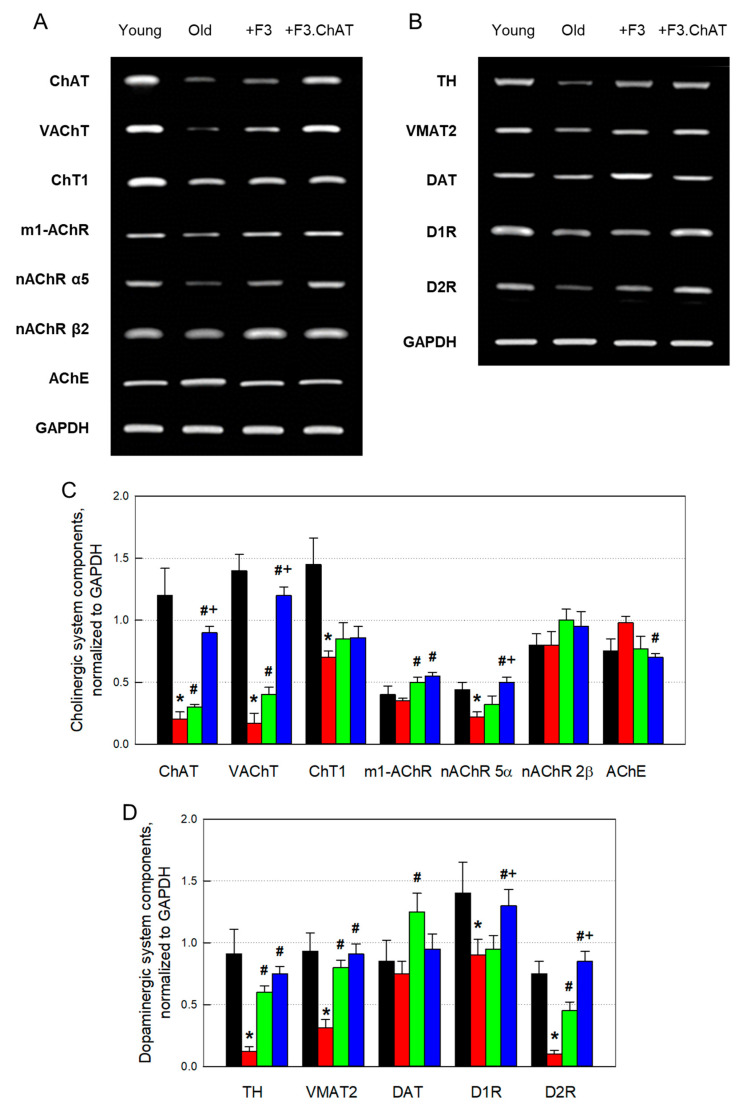
Restorations of cholinergic ((**A**,**C**), *n* = 8–10/group) and dopaminergic ((**B**,**D**), *n* = 8–10/group) nervous systems following F3 or F3.ChAT cell transplantation. RT-PCR (**A**,**B**) and quantitative analyses (**C**,**D**) of the mRNA expression of cholinergic and dopaminergic components. Black: young animals (7 weeks old); red: aged animals; green: aged animals transplanted with F3 cells (1 × 10^6^ cells every month); blue: aged animals transplanted with F3.ChAT cells (1 × 10^6^ cells every month). ChAT: choline acetyltransferase. VAChT: vesicular acetylcholine transporter. ChT1: choline transporter 1, m1-AChR: muscarinic 1 acetylcholine receptor. nAChR: nicotinic acetylcholine receptor. AChE: acetylcholinesterase. GAPDH: glyceraldehyde-3-phosphate dehydrogenase. TH: tyrosine hydroxylase, VMAT2: vesicular monoamine transporter 2. DAT: dopamine transporter, DR: dopamine receptor. * Significantly different from young rats (*p* < 0.05). # Significantly different from aged rats (*p* < 0.05). + Significantly different from F3 treatment (*p* < 0.05).

**Figure 5 cells-12-02711-f005:**
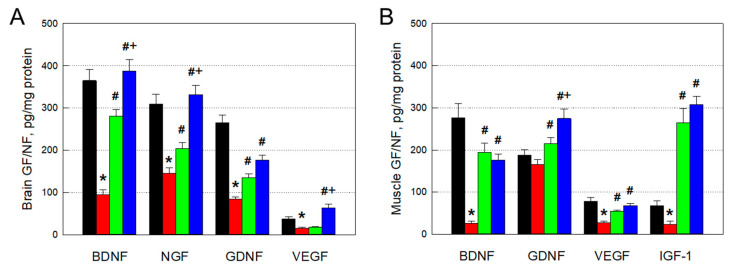
Restoration of growth factors (GFs) and neurotrophic factors (NFs) in the brain ((**A**), *n* = 8–10/group) and muscles ((**B**), *n* = 8–10/group). Black: young animals (7 weeks old); red: aged animals; green: aged animals transplanted with F3 cells (1 × 10^6^ cells every month); blue: aged animals transplanted with F3.ChAT cells (1 × 10^6^ cells every month). BDNF: brain-derived neurotrophic factor. NGF: nerve growth factor. GDNF: glial cell-derived neurotrophic factor. VEGF: vascular endothelial growth factor. IGF-1: insulin-like growth factor-1. * Significantly different from young rats (*p* < 0.05). # Significantly different from aged rats (*p* < 0.05). + Significantly different from F3 treatment (*p* < 0.05).

**Figure 6 cells-12-02711-f006:**
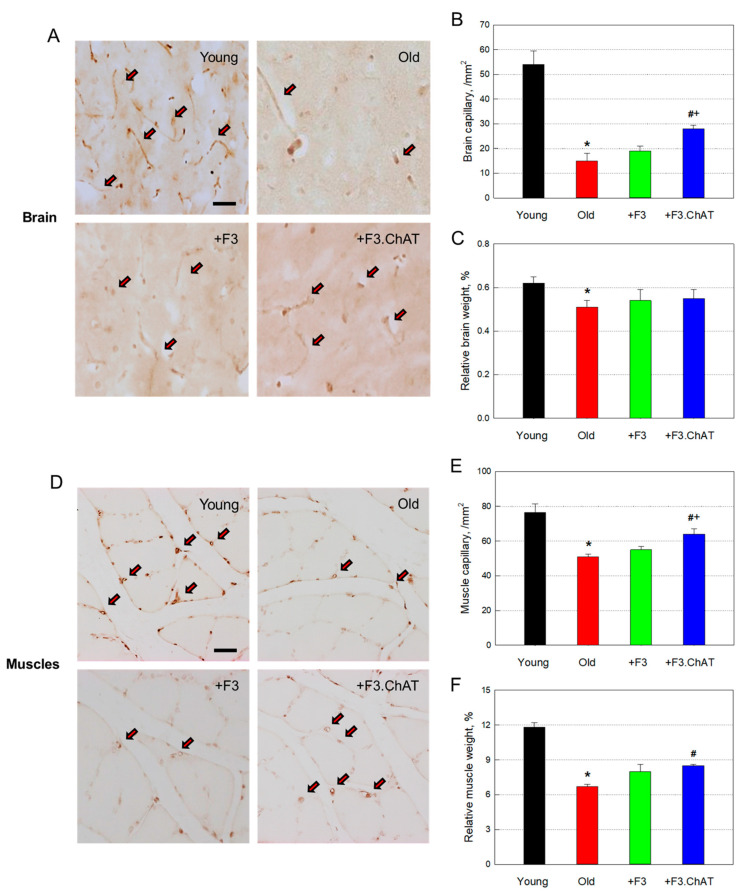
Microvessel (vWF-positive) density in the brain ((**A**,**B**), *n* = 8–10/group) and muscles ((**D**,**E**), *n* = 8–10/group) and relative brain (**C**) and muscle (**F**) weights. Arrow: vWF-positive blood vessel. Sale bar = 50 μm. * Significantly different from young rats (*p* < 0.05). # Significantly different from aged rats (*p* < 0.05). + Significantly different from F3 treatment (*p* < 0.05).

**Figure 7 cells-12-02711-f007:**
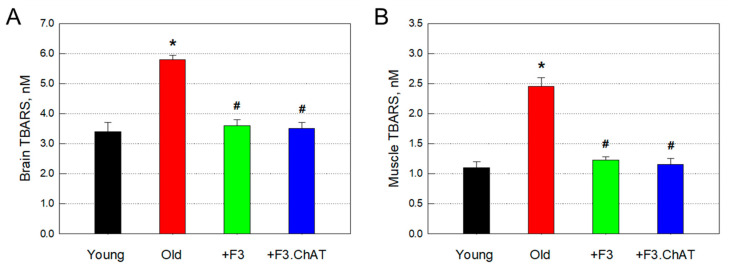
Concentration of thiobarbituric acid-reactive substances (TBARS) in the brain ((**A**), *n* = 8–10/group) and muscles ((**B**), *n* = 8–10/group). * Significantly different from young rats (*p* < 0.05). # Significantly different from aged rats (*p* < 0.05).

**Figure 8 cells-12-02711-f008:**
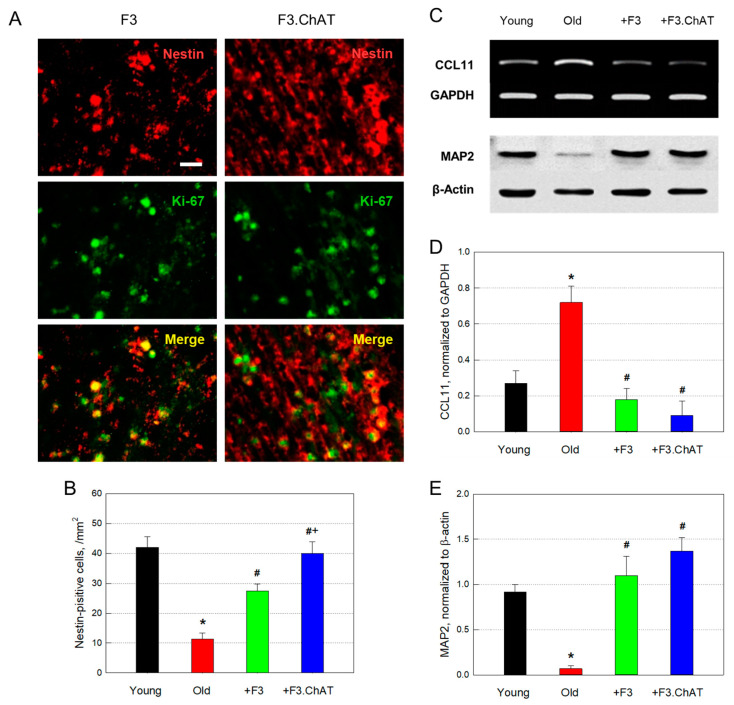
Host neural stem cell (nestin-positive: red) regeneration and proliferation (Ki-67-positive: green) ((**A**,**B**), *n* = 5/group); CCL11 mRNA downregulation ((**C**,**D**), *n* = 8–10/group) and neuronal integrity (MAP2) restoration ((**C**,**E**), *n* = 8–10/group). (**A**): immunohistochemical staining, (**B**): RT-PCR analysis, (**C**): western blot analysis. Double-positive cells (yellow in Merge) in A indicate proliferating neural stem cells. Sale bar = 20 μm. * Significantly different from young rats (*p* < 0.05). # Significantly different from aged rats (*p* < 0.05). + Significantly different from F3 treatment (*p* < 0.05).

## Data Availability

Data are unavailable due to ethical restrictions.

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
