# Peer review of "Repeated Intravenous Administration of Human Neural Stem Cells Producing Choline Acetyltransferase Exerts Anti-Aging Effects in Male F344 Rats"

_cells, 2023, doi:10.3390/cells12232711_

Round 1
Reviewer 1 Report (New Reviewer)
Comments and Suggestions for Authors
This manuscript investigates how intracerebroventricular transplnatation human neural stem cells expressing choline acetyltransferase affects learning and memory in a rat model. The results are of value to others in the field. I do have some suggestions for the authors to consider when this manuscript is revised.
1) Transcriplantation of human neural stem cells to a rat model may trigger immune rejection, so would it better to use rat neural stem cells instead?
2) As for expression, mRNA has been detected, but it is important that the expression is indeed enhanced at the protein level by using immunoblotting (to complement immunohistochemical staining.
Comments on the Quality of English Language3) The manuscript needs some editing. For example, there are too many acronyms in the abstract, which makes the readability an issue. Also, word "gene" can be removed from the title and also from the following sentence in the abstract "...over-expressing choline acetyltransferase (F3.ChAT) gene, an enzyme responsible for...".
Author Response
Please see the attached file (Answers to reviewers' comments).
Thank you!

Reviewer 2 Report (New Reviewer)
Comments and Suggestions for Authors
This is a longevity study, one of many over the past year or so. Given the repeated reports on the decline of cholinergic signaling with age as a threat to cognition, alertness and longevity, the authors engineered cells expressing choline acetyl transferase (ChAT) as a hallmark of cholinergic signaling and implanted them into the brain of aging rodents, an original approach that deserves appreciation. The animals showed better cognition and energy, presented revived physical activities and seemed re-juvenile in general, which are impressive messages of this manuscript. However, there are less-exciting aspects as well, which call for revision to improve the message conveyed by this article and enhance its cutability, as is listed below.
First, this is a very long and detailed manuscript, especially considering the length of competing ones; reporting, for example, how exchanging the cerebrospinal fluid with juvenile one improves longevity. The authors are encouraged to remember that this is one of a series of multi-discipline studies on how to fight aging; some do it by injected cerebrospinal fluid from young animals, and cholinergic cells that are injected to aged rats may be considered as part of this series.
Second, while the findings are convincing, this entire study was done only in male rats, which is a half job; repeating the same tests in female rats can teach us a lot and send an encouraging message to other scientists to do the same. This is particularly important given the recent study that reported loss of mitochondrial small RNA fragments that regulate cholinergic features in aged humans, but only in females (Shulman et al., Alzheimer&Dementia 2023) which should be cited here.
Third, the paper is very long, partially due to the phrasing- the authors are encouraged to remember that whatever was published already is considered to be the scientific truth, while whatever is currently presented should be in past tense; and there is no need to say ‘studies have shown that’… when citing others.
Fourth, comparison to others’ studies on aging reversal is missing, starting with extracts of tissues and proceeding to replacement of the CSF; and there is no mention at all of RNA regulators as being involved in the presented anti-aging process. Summarizing what had already been done in a presented field of research is a must.
Last, but not least, the language can and should be improved. Assisted by a native English speaker, the authors are encouraged to shorten the text, cite others who also improved aging and cover the reports on cholinergic decline under aging which would fit here perfectly.
Comments on the Quality of English LanguageAssisted by a native English speaker, the authors are encouraged to shorten the text.
Author Response
Please see the attached file (Answers to reviewers' comments).
Thank you!

Reviewer 3 Report (New Reviewer)
Comments and Suggestions for Authors
Overall, this manuscript is well written and documented, and the conclusions are supported by the data presented.
Author Response
Please see the attached file (Answers to reviewers' comments).
Thank you!
This manuscript is a resubmission of an earlier submission. The following is a list of the peer review reports and author responses from that submission.
Round 1
Reviewer 1 Report
Comments and Suggestions for Authors
1) Introduction is too complex to understand, make it more streamlined with the study
2) there is no mention of the N number, please specify the no of animals used, and how many times the experiment was done and also label all the graphs properly with each color denotation.
3) Fig 1 A: a) how did you distinguish between the slow and fast-moving, give some reference points or cite papers that have done similar. b) All labels and N
4) Fig2: N missing and also labels of each color
5) Fig 3: Scale bar on the image, N and also significance is missing in the experiment.
6) Fig 4: Quantification of western blot needed to say what significantly increased or decreased.
7) Fig5: Scale bar missing and also labels.
8) Fig 7: Which condition is shown in the images? Scale bar missing? N number is not there. Quantification of western blot images not there.
9) Discussion is too complicated to understand, simplify the language.
Comments on the Quality of English LanguageThe text is very hard to understand. Even the title of this paper is too wordy to understand. Revise the paper in a simple way, rather than long sentences convey the idea in easy-to-understand sentences.
Reviewer 2 Report
Comments and Suggestions for Authors
It is well known that as we age, our brains experience shrinkage and changes in neurotransmitter levels, resulting among others in a decline in cognitive and motor function. These changes are caused by deficits in acetylcholine and dopamine. Decreased dopaminergic function has a particularly strong impact on cognitive flexibility. In this study, the potential of neuronal stem cells for anti-aging therapy was investigated. Intravenous injections of neural stem cells were administered to aging rats between 10 and 21 months of age. Monthly injections of neural stem cells, specifically those with overexpression of the choline acetyltransferase gene, improved motor balance, coordination, and cognitive function, as demonstrated by behavioral tests such as rota-rod, passive avoidance, and Morris water maze tests. Comprehensive postmortem analyses using relevant methods showed that cell transplantation prevented age-related decreases in acetylcholine in CSF and muscle, as well as changes in cerebral expressions of functional genes of cholinergic and dopaminergic systems (related to synthesis, transport, and receptors). Additionally, microvessel density reduction, oxidative damage to the brain, and brain atrophy were prevented. Immunohistochemistry with human mitochondria biomarkers confirmed that intravenously transplanted cells were found in the hippocampus and cortex, proving they had crossed the blood-brain barrier.
The data is presented clearly, the discussion is sufficient and the references are well chosen.
What we now look forward to is a more deep exploration of the interaction between the host brain and transplanted cells enabling the fast development of such innovative regenerative therapy